# Phage-encoded bismuth bicycles enable instant access to targeted bioactive peptides
Sven Ullrich [1,2], Upamali Somathilake [1,2], Minghao Shang [1] & Christoph Nitsche [1] ✉

Genetically encoded libraries play a crucial role in discovering structurally rigid, high-affinity macrocyclic peptide ligands for therapeutic applications. Bicyclic peptides with metal centres like bismuth were recently developed as a new type of constrained peptide with notable affinity, stability and membrane permeability. This study represents the genetic encoding of peptide-bismuth and peptide-arsenic bicycles in phage display. We introduce bismuth tripotassium dicitrate (gastrodenol) as a water-soluble bismuth(III) reagent for phage library modification and in situ bicyclic peptide preparation, eliminating the need for organic co-solvents. Additionally, we explore arsenic(III) as an alternative thiophilic element that is used analogously to our previously introduced bicyclic peptides with a bismuth core. The modification of phage libraries and peptides with these elements is instantaneous and entirely biocompatible, offering an advantage over conventional alkylation-based methods. In a pilot display screening campaign aimed at identifying ligands for the biotin-binding protein streptavidin, we demonstrate the enrichment of bicyclic peptides with dissociation constants two orders of magnitude lower than those of their linear counterparts, underscoring the impact of structural constraint on binding affinity.

Genetically encoded libraries are key technologies in peptide drug discovery[1–3]. Integrated into display screening platforms, their vast sequence diversity facilitates the identification of peptide hits for therapeutically relevant targets[4–6]. Chemical library modifications compatible with the biological environment of these screening platforms have been shown to enhance the pharmaceutical properties of the displayed peptides[7–9]. Particularly desirable are strategies that constrain the entire peptide structure, such as macrocyclisation or bicyclisation[9–11]. Restricting the conformational freedom of a peptide can cause dramatic improvement in bioactivity, stability, and membrane permeability[9,12,13]. Hence, constrained peptides derived from genetically encoded library selections are anticipated to initiate a wave of peptide therapeutics[3,14–16].

Especially the strong constraint of peptide multicycles has made them an increasingly attractive modality of therapeutic candidates[11,14,16–20]. In this context, our group has recently introduced peptide–bismuth bicycles[21,22]. With the peptide component connected to a trivalent bismuth core via three sulfur atoms from cysteines, these bicycles feature a unique architecture[21]. Importantly, the formation of peptide–bismuth bicycles occurs instantaneously and can be initiated in situ[21]. The bismuth core of these bicycles also holds promise for future cancer therapeutics, as the radioisotope bismuth-213 is an important emitter in targeted alpha therapy[23,24]. To allow for the rapid identification of peptide–bismuth bicycles for virtually any given target, we set out to investigate the compatibility of bismuth bicyclisation with a genetically encoded library.

Phage display is a versatile screening technique with extensive application in antibody development and drug discovery[25–27]. It typically requires the presentation of peptides on the M13 bacteriophage surface, e.g., on the coat proteins pIII or pVIII (Fig. 1a), which is facilitated by the integration of a semi-randomised library into the phage genome[25–27]. Chemical modification of the displayed peptides expands library diversity beyond the standard genetic code, enabling the selection of peptide ligands with enhanced affinity and stability, including macrocyclic peptides[7,14,16,28]. Conventional approaches for the display of bicyclic peptide libraries on phage centre on the modification of three-cysteine residues using alkylating agents[7,29]. Most commonly, 1,3,5-tris(bromomethyl)benzene (TBMB) is used (Fig. 1b)[29]. While TBMB and its derivatives have contributed substantially to the identification of bicyclic peptide ligands for various protein targets[14], it has minor shortcomings such as cross-reactivity with other nucleophiles in peptides or proteins, incompatibility with common reducing agents, and comparatively harsh reaction conditions involving organic co-solvent (Fig. 1b)[29,30].

[1]Research School of Chemistry, Australian National University, Canberra ACT 2601, Australia. [2]These authors contributed equally: Sven Ullrich, Upamali Somathilake. ✉e-mail: christoph.nitsche@anu.edu.au

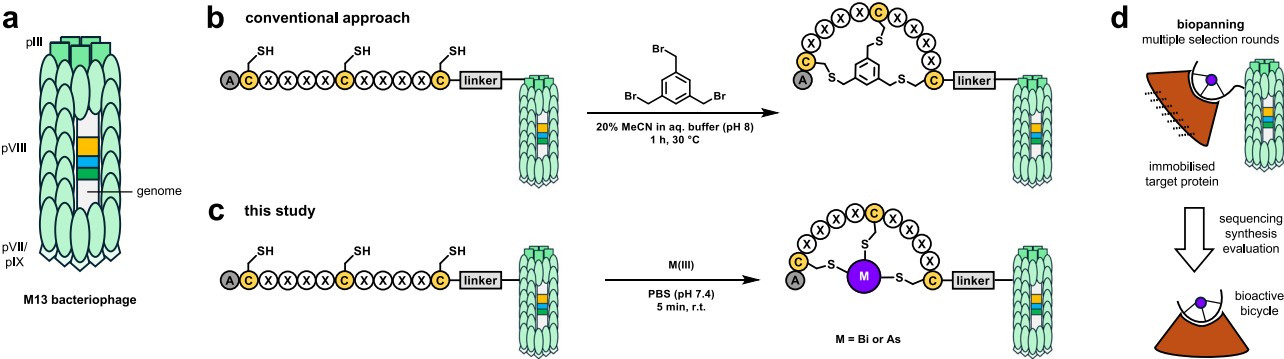

**Fig. 1 | Strategies for the display of bicyclic peptides on phage. a** Schematic representation of the M13 bacteriophage structure used in this study (green). **b** Conventional approach to display bicyclic peptide libraries on the coat protein of the M13 phage. **c** Our strategy is to display bicyclic peptide libraries on the coat protein of the M13 phage using thiophilic metal(loid)s like bismuth(III) or arsenic(III) (purple). Note that the choice of coat protein (e.g., pIII or pVIII) is not essential for either modification strategy. **d** Panning and selection of modified phage libraries against protein targets (brown) to identify bioactive bicyclic peptides with a metal(loid) core.

In contrast to TBMB, the reaction of three cysteines residues in a peptide with bismuth(III) is instantaneous, selective, and proceeds in an aqueous buffer at room temperature and physiological pH[21]. We, therefore, hypothesised that the strong selectivity and biocompatibility of bismuth-based peptide bicyclisation would be a valuable technique in the context of genetically encoded libraries, where it could also mitigate the minor challenges of commonly used alkylating agents. In addition, we demonstrate that the concept of peptide bicycle formation by bismuth can be extended to other thiophilic metals and metalloids of interest (Fig. 1c) and applied to phage display (Fig. 1d).

The present study was published as a preprint[31]. A related study by Kong and co-workers describing the phage display of peptide–bismuth bicycles was also recently published[32]. Following the modification of a three-cysteine phage library with BiBr$_3$ from acetonitrile, the group identified peptide–bismuth bicycles targeting maltose-binding protein in a display screening. One bicyclic peptide showed two orders of magnitude improved affinity towards its target compared to its linear counterpart. Both independent studies are complementary in demonstrating the ease and robustness of this method.

## Results

### Exploration of bismuth(III) and arsenic(III) reagents for phage library modification

We chose the commonly used M13 phage for our investigations and selected the pVIII coat protein for bicyclic peptide display (Fig. 1a). A semi-randomised library containing an AC$\underline{X_4}$C$X_4$$\underline{C}$GGGGENLYFQS extension (where X is any canonical amino acid) at the N-terminus of pVIII formed the basis of our studies. This construct comprises a glycine linker and TEV protease recognition site between the displayed peptide and pVIII to enable selective peptide release from the phage if required. The equal spacing of cysteine residues in the construct is reminiscent of the libraries used for modification with TBMB and our previously published peptide–bismuth bicycles[21,22,29].

Our initial efforts were focused on potential 'scaffolding reagents'. We selected bismuth tribromide (BiBr$_3$) to replicate the conditions used for bismuth bicyclisation of peptides, as shown in our previous studies[21,22]. Given that BiBr$_3$ and most other Bi(III) salts are insoluble in water at near-neutral pH, concentrated DMSO stocks of bismuth salt are typically necessary for peptide modification in buffer solutions. Anticipating potential challenges associated with organic co-solvent when applied to phages, we explored the fully buffer-soluble bismuth tripotassium dicitrate (gastrodenol) as an alternative. Consequently, we employed both bismuth reagents to assess their efficacy in phage display. Lastly, we envisioned As(III) from water-soluble sodium arsenite (NaAsO$_2$) as a substitute for Bi(III) in library modification and used it analogously for phage library modification.

In preparation for the phage library bicyclisation, we verified that our bismuth bicyclisation is amenable to protein modification under fully biocompatible conditions. A three-cysteine peptide resembling our library (AC$\underline{GGSG}$C$\underline{GGSG}$C$\underline{GGG}$ENLYFQS) was fused to the N-terminus of GB1 (domain B1 of immunoglobulin binding protein G from *Streptococcus*) as model system, forming the bismuth-bicycle conjugate with gastrodenol in PBS (pH 7.4) after 5 min (Supplementary Fig. 19, Supplementary Tables 8 and 9). We also found that phage infectivity is uncompromised by the addition of 120 μM of As(III) or Bi(III) for 5 min in a plaque-forming assay (Supplementary Fig. 20), allowing us to proceed with these conditions.

### Display screenings of bismuth- and arsenic-bicyclised phage libraries

For the display screenings, we used the biotin-binding protein streptavidin as the model target. Our libraries were modified to bicyclic peptide phage libraries (Supplementary Fig. 1) and underwent four rounds of biopanning against the immobilised target. To bias the selection for peptides that bind to the ligand-binding site, we conducted competitive elution with biotin in each round of the selection process (Fig. 2a). In addition to both Bi(III) screenings (BiBr$_3$ and gastrodenol), we performed two As(III) screening replicates with NaAsO$_2$ in order to assess the robustness and reproducibility of our strategy. Gratifyingly, both Bi(III) screenings featured similarly enriched peptides despite the differences in the modification reagent (Fig. 2b, c). The two independently conducted As(III) screenings also enriched identical top sequences in different proportions, indicating excellent replicability (Fig. 2d, e). Bi(III) and As(III) selections yielded distinct hits in terms of the strongest enriched sequences, although several of these appeared at lower frequencies in the Bi(III) and As(III) deep sequencing datasets (Supplementary Figs. 6–9, Supplementary Data 1). Many top hits contained an HPQ/M motif, which has previously been described to facilitate peptide interactions with the biotin-binding site of streptavidin[33–36]. A few enriched sequences without this motif are known to us from other screenings as suspected non-target binders. An analysis of the most frequently occurring dipeptide motifs[37] further confirmed the pronounced appearance of HP and PQ/PM within the top 25 hit sequences (Fig. 2b–e).

### Synthesis and evaluation of peptide–bismuth and peptide-arsenic bicycles

We synthesised four linear peptide sequences (**1–4**; H-AC$\underline{X_4}$C$X_4$$\underline{C}$-NH$_2$) discovered in the Bi(III) and As(III) selection campaigns against streptavidin using automated solid-phase peptide synthesis (SPPS) on Rink amide resin (Supplementary Table 5). The purified peptides were bicyclised with water-soluble Bi(III) from gastrodenol and As(III) from NaAsO$_2$ (Fig. 3a). Bicyclisation is instantaneous and quantitative in buffer at physiological pH

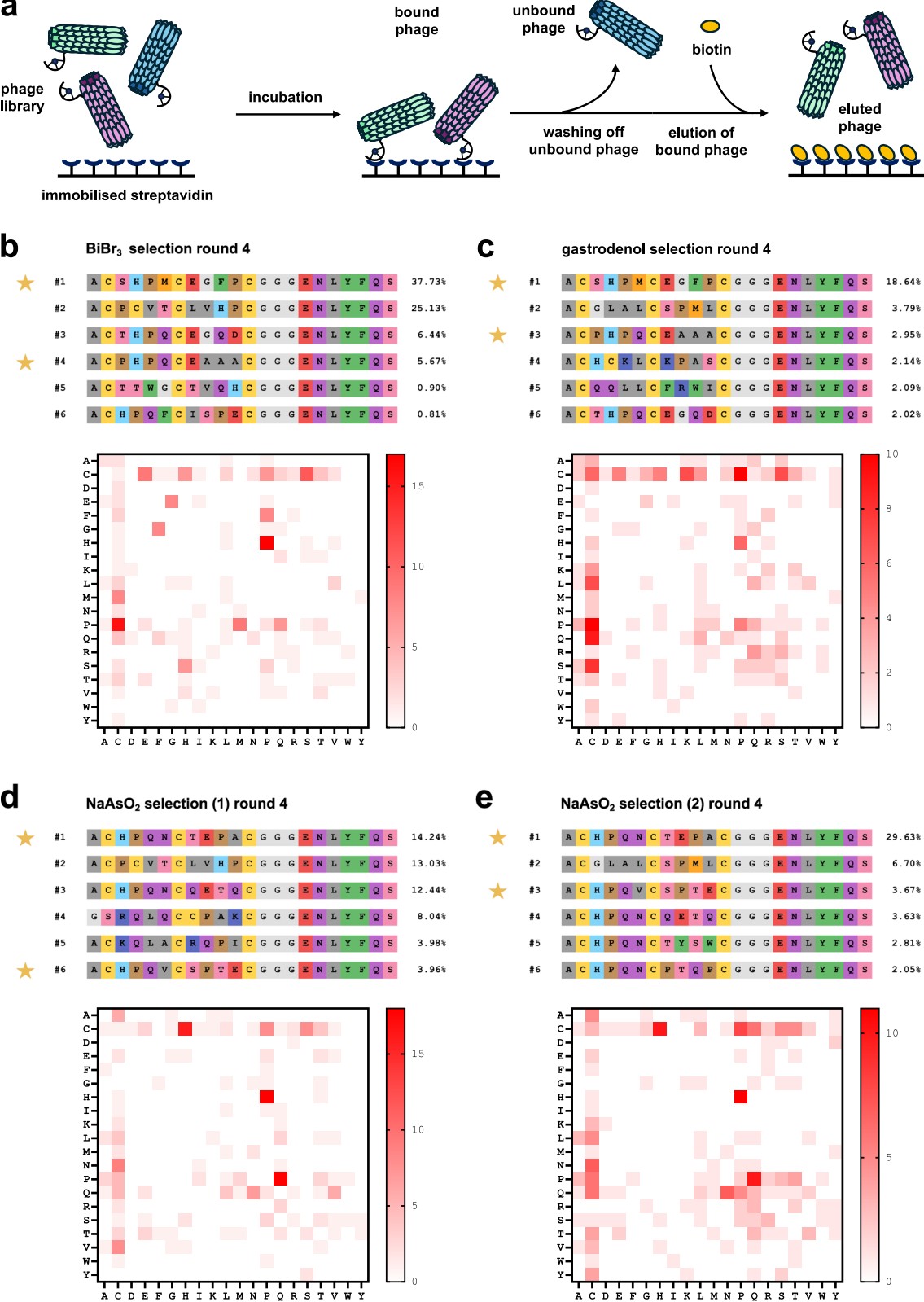

**Fig. 2 | Screening and enrichment of bicyclic peptides binding to streptavidin.**
**a** Biopanning of a bicyclic peptide phage library (blue, green, violet) against immobilised streptavidin using competitive elution with biotin (yellow). **b–e** Top six hits identified by Nanopore sequencing, including dipeptide motif analysis of the top 25 hits of round 4 from **b** BiBr$_3$, **c** gastrodenol, **d** and **e** NaAsO$_2$ selection campaigns. Identified top six sequences are indicated with their rank and percent enrichment.

The dipeptide motif analysis shows the occurrence of different pairs of amino acids (vertical: first amino acid; horizontal: second amino acid) within the cysteine-flanked randomised parts (CX$_4$CX$_4$C) of the top 25 peptide insert hits (Supplementary Figs. 6–9). Peptide sequences selected for synthesis and further evaluation are highlighted with a golden star (Supplementary Table 5).

(Fig. 3b). This in situ modification facilitated the direct evaluation of bicyclic peptides by SPR against immobilised streptavidin, eliminating the need for laborious purification of the bicyclic peptide.

All four tested top enriched bicyclic peptides (1b, 2b, 3a, 4a) exhibited micromolar affinity towards streptavidin (Fig. 4a, b), with peptide 2b from the Bi(III) screening campaign standing out as the best of the four analysed peptides ($K_D$ = 7.4 µM). We selected the two peptides with the highest

affinity from the bismuth (2b) and arsenic (3a, $K_D$ = 11.7 µM) series and compared them to their linear analogues (Fig. 4c). Importantly, in both cases, the linear unmodified peptide exhibited a significantly lower affinity for streptavidin. Arsenic bicycle 3a is approximately 80 times more active than its linear counterpart 3, while bismuth bicycle 2b is even approximately 200 times more active than its linear precursor 2. We also generated peptide–bismuth bicycles 3b and 4b to evaluate sequences originating from

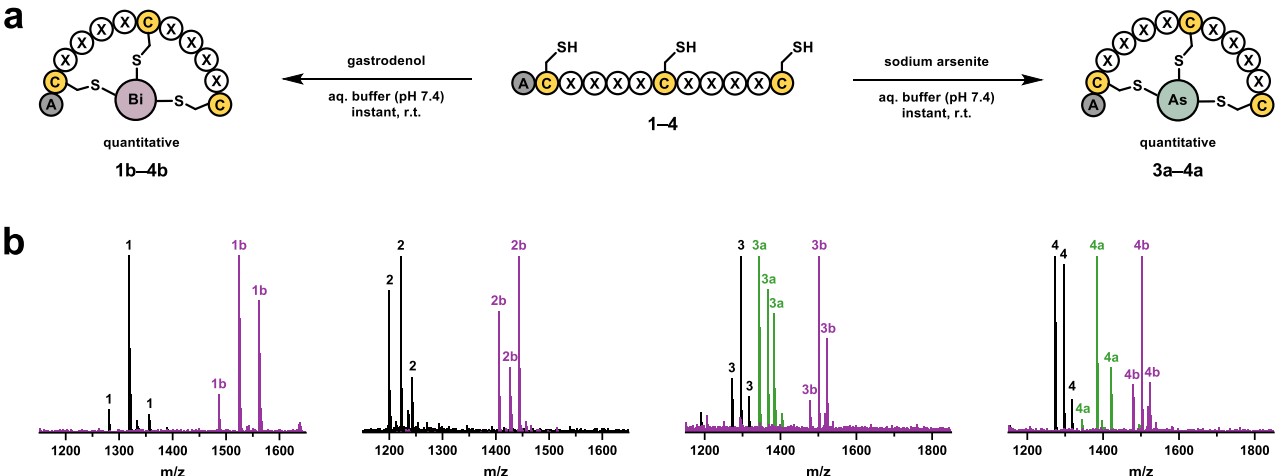

**Fig. 3 | Instant and quantitative peptide bicyclisation using bismuth and arsenic reagents. a** Conversion of synthetic linear peptides from the streptavidin screening campaigns into peptide–bismuth bicycles (purple) using gastrodenol (bismuth tripotassium dicitrate) or peptide-arsenic bicycles (green) using sodium arsenite (NaAsO$_2$). **b** High-resolution mass spectrometry overlays of linear peptides (1–4)

and the respective bicycles (1b–4b, 3a–4a) indicate their quantitative in situ conversion. Peaks corresponding to the ESI-MS adducts of the compounds are labelled with their compound code. Details regarding expected and observed masses are provided in Supplementary Figs. 10–12 and Supplementary Tables 6, 7.

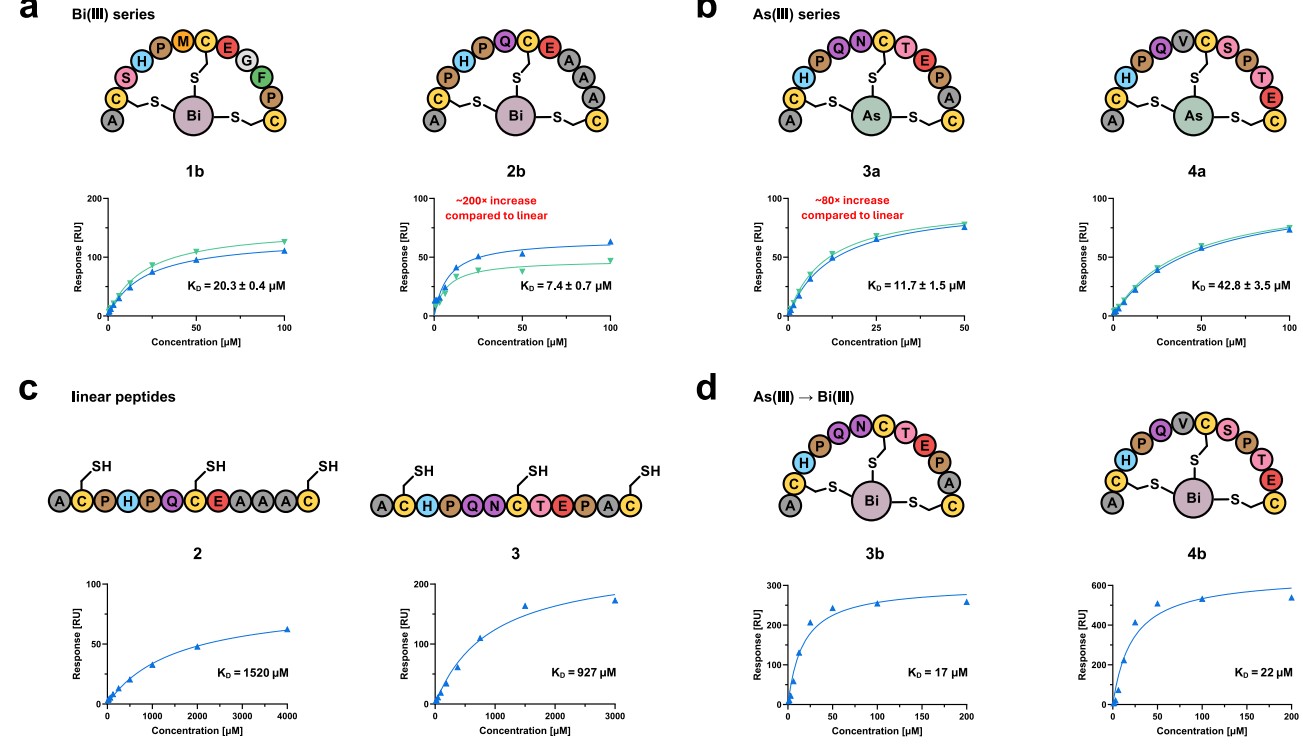

**Fig. 4 | SPR evaluation of binding interactions of selected peptides with streptavidin. a** Bicycles identified from the bismuth screenings (1b, 2b). **b** Bicycles identified from the arsenic screenings (3a, 4a). **c** Linear peptides (2, 3) without metal(loid) for comparison. **d** Top enriched sequences from the arsenic screenings evaluated as their respective bismuth analogues (3b, 4b). Bicyclic peptides 1b, 2b, 3a,

and 4a were tested in duplicate (n = 2; green, blue) in separate flow-cells. Linear peptides 2 and 3 and bicyclic peptides 3b and 4b were evaluated in one single experiment (n = 1; blue). $K_D$ values are shown within the binding curves (± SD where applicable). Note that RU$_{max}$ differences can stem from independent replication within separate flow cells with slightly varying protein immobilisation levels.

the As(III) screenings with a Bi(III) core (Fig. 3b). Both sequences also appear to be slightly enriched in the Bi(III) screenings (Supplementary Table 4). SPR analysis showed that both peptide–bismuth bicycles **3b** and **4b** exhibited two-digit micromolar dissociation constants against streptavidin (Fig. 4d).

## Discussion

As recently demonstrated, linear peptides containing three-cysteine residues can be modified by Bi(III) to generate constrained bicyclic peptides under biocompatible conditions[21,22]. Our previous studies relied on water-insoluble BiBr$_3$ dissolved in DMSO, which may cause complications. This study introduces the water-soluble pharmaceutical drug gastrodenol (bismuth tripotassium dicitrate) as an alternative. Gastrodenol serves as an easily accessible and buffer-soluble reagent for the quantitative generation of bismuth-peptide bicycles from linear precursors in buffer at physiological pH. Additionally, we induced bicycle formation by As(III) using water-soluble NaAsO$_2$ (Fig. 3). While the generation of arsenic-peptide bicycles might be of limited interest for applications in biomedical research due to their probable toxicity, this example demonstrates the potential of expanding our macrocyclisation strategy to other thiophilic metal(loid)s for theranostics. Modification of three-cysteine phage libraries by gastrodenol and NaAsO$_2$ offers notable advantages over conventional modification with TBMB because these reagents (i) eliminate the use of organic co-solvent, (ii) react instantaneously at room temperature, (iii) tolerate the reducing agent TCEP, and (iv) exhibit strong selectivity for the three cysteines residues.

We validated our methodology with screening campaigns against streptavidin as a model target (Fig. 2a). The remarkably strong interaction between streptavidin and biotin is extensively studied[38] and peptides binding to the biotin-binding site have been previously identified using phage display and alternative technologies[33–36,39–43]. As a result, it is well-known that the motif HPQ/M is highly conserved in peptides binding to the biotin-binding site[35,36,44]. Crystal structures with HPQ peptides are even documented (e.g., PDB: 1SLG, 1SLD)[35,45,46]. Competitive elution of phage with biotin ensured that most eluted phages bound to the biotin-binding site. Consequently, all four of our screening campaigns displayed strong enrichment of the HPQ/M motif, as demonstrated by the dipeptide motif analyses conducted for each library (Fig. 2b–e). Interestingly, both Bi(III) libraries exhibited the highest enrichment for an HPM peptide, while both As(III) libraries showed the highest enrichment for an HPQ peptide. The vast majority of the HPQ/M motifs appeared in the N-terminal peptide loop of enriched sequences (Supplementary Fig. 6–9, Supplementary Data 1), which might be related to steric effects between target and phage protein during screening. The fact that both arsenic and bismuth screening pairs enriched identical number-1 hits underscores the robustness and effectiveness of the method. However, it appears that the most enriched sequences in the bismuth screenings are distinct from those in the arsenic screenings, which might suggest sensitivity to the geometry of the linker atom. As the difference in the van der Waals radii between arsenic and bismuth is small[47], this effect might be primarily mediated by variations in metal(loid)–sulfur bond angles and lengths. It is also noteworthy that the sequences enriched in the arsenic screenings (e.g., **3** and **4**) also appear in the bismuth screenings, however, at lower abundance (Supplementary Table 4). Two peptides enriched in the arsenic screening and tested as their bismuth analogues indicate similar affinity towards streptavidin (Fig. 4).

All selected bicyclic peptides originating from the screening campaigns showed an affinity for streptavidin in the low micromolar range as investigated by SPR, which aligns with previously reported peptides incorporating the HPQ motif[35,48]. More important than the absolute affinities is the observation that the linear analogues (**2** and **3**) exhibit two orders of magnitude lower affinity for streptavidin than their bicycles (**2a** and **3b**), despite the presence of the identical HPQ motif (Fig. 4). This observation can be attributed to an entropic effect related to the preorganisation of the bicycle compared to the linear peptide, as has been previously reported for peptide substrates of flaviviral proteases assessed in presence and absence of Bi(III)[21,49].

The affinity increase from linear to bicyclic peptide was also observed by Kong and co-workers for their main peptide sequence targeting maltose-binding protein[32]. In contrast to our study, this group chose to display their three-cysteine peptides on pIII and used only BiBr$_3$ from acetonitrile for library bicyclisation. Both independent studies underscore the high relevance of using thiophilic elements like bismuth in phage display to gain access to targeted bioactive peptide bicycles with unique architectures.

## Conclusion

We present genetically encoded bicyclic peptide libraries containing bismuth or arsenic atoms in the peptide core. This method complements existing and well-established strategies using alkylating agents like TBMB. Bicycles form instantaneously with Bi(III), As(III) and potentially alternative thiophilic metal(loid) reagents in buffer at physiological pH. The use of water-soluble gastrodenol eliminates the need for previously necessary organic co-solvents. Modification of semi-randomised phage libraries proves highly reliable, as demonstrated by our similar results from repeated screenings. Bicyclic peptides exhibit dramatically increased affinity to the protein target compared to their linear analogues. We anticipate that this strategy will find broad applications, such as in the identification of bicyclic peptide ligands for radiopharmaceuticals.

## Methods

### Design of phage-displayed library

The naïve unmodified M13 phage library used in this study was a gift from Prof. Ratmir Derda (University of Alberta). It encodes an ACX$_4$CX$_4$CGGGENLYFQS peptide at the N-terminus of recombinant pVIII on a type 88 phage vector that is similar to previously published work[50].

### Modification of phage-displayed libraries

Amplified phage libraries were reduced using 2 mM TCEP-NaOH in sterile PBS (pH 7.4) for 30 min at room temperature. Using spin desalting columns, residual TCEP was removed via centrifugation at $2000 \times g$ for 2.5 min. The reduced phage library solutions were then modified by exposure to either BiBr$_3$ (50 mM in DMSO), gastrodenol or NaAsO$_2$ (both 50 mM in sterile ultrapure water) at final concentrations of 120 µM for 5 min, before excess reagent was removed using spin desalting columns ($2000 \times g$, 2.5 min).

### Biopanning of modified libraries

Biopanning campaigns were conducted against streptavidin-coated magnetic particles. Four selection rounds with increasing stringency were performed for each modified phage library (Supplementary Table 1). For selection, a competitive elution protocol using 0.1 mM biotin was employed. Sanger (Supplementary Figs. 2–5) and Nanopore (Supplementary Figs. 6–9) sequencing were used to analyse the eluted phages (Supplementary Tables 2, 3). A portion of the remaining eluted phage was amplified in *Escherichia coli* ER2738 for subsequent selection rounds.

### In situ peptide bicyclisation

Purified linear peptide was dissolved in aqueous reducing buffer (20 mM HEPES–KOH pH 7.4, 150 mM NaCl, 2.5 mM TCEP) and exposed to 1.1 equivalents of gastrodenol or sodium arsenite at 1 mM, while the mixture was gently vortexed. Quantitative conversions of the linear peptides to the bicycles were confirmed with high-resolution mass spectrometry (Supplementary Figs. 10–12) and liquid chromatography–mass spectrometry (Supplementary Figs. 13–16).

### Surface plasmon resonance

SPR measurements were performed with 20 mM HEPES–KOH pH 7.4, 150 mM NaCl, 0.5 mM TCEP, 0.05% (v/v) polysorbate 20 as running buffer. For each experiment, 100–150 µg streptavidin in 10 mM sodium acetate pH 3.9 buffer was freshly immobilised on a sensor chip flow-cell using EDC/NHS chemistry at 25 °C in absence of TCEP. All binding experiments were

conducted at 20 °C in single-cycle kinetics mode (Fig. 4, Supplementary Figs. 17, 18, Supplementary Data 2).

## Reporting summary

Further information on research design is available in the Nature Portfolio Reporting Summary linked to this article.

## Data availability

Further details on the materials and methods are presented in the Supplementary Information. The article and Supplementary Information contain all data necessary to support the findings and conclusion of this study. Supplementary Data 1 and 2 contain deep sequencing results and SPR data, respectively. Additional data are available from the corresponding author upon reasonable request.

## Code availability

Next-generation sequencing data was analysed with a Python script to extract, rank and visualise enriched sequences from the display screening, which is available as Supplementary Software 1.

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

## Acknowledgements
We thank the Australian Research Council for funding support, including a Discovery Project (DP230100079) and a Future Fellowship (FT220100010). We acknowledge Prof. Ratmir Derda (University of Alberta) for generously providing training and the unmodified phage library. We also thank Prof. Derda's group members, Dr. Alexey Atrazhev, for sharing his expertise in cloning and Dr. Arunika Ekanayake and Kejia Yan, for their valuable discussions. We also thank Dr. Claudia Yan at the Biomolecular Resource Facility (Australian National University) for performing Sanger sequencing, and Lavi Singh and Lydia Murphy in the group of A/Prof. Benjamin Schwessinger (Australian National University) for performing and supporting Nanopore sequencing. We further acknowledge Dr. Shouvik Aditya (Australian National University) for his assistance in SPR training and analysis, as well as Richard Morewood (Australian National University) for training and assistance in HPLC purification.

## Author contributions
S.U. and C.N. wrote the paper with contributions from all authors. C.N. designed the research project. S.U. and U.S. amplified and modified phage libraries. U.S. performed all phage viability and selection experiments. S.U. and U.S. prepared phage library samples for sequencing. S.U. evaluated deep sequencing data with custom code. S.U. and M.S. conducted peptide synthesis and analysis. M.S. purified all linear peptides. S.U. produced all bicyclic peptides. S.U. performed all SPR experiments. S.U., U.S. and C.N. prepared the graphics. All authors reviewed and approved the final manuscript.

## Competing interests
The authors declare no competing interests.
