## [Peer Review File · Communications Chemistry]

Reviewers' comments:

Reviewer #1 (Remarks to the Author):

This contribution describes the development and proof-of-concept application of phage libraries of bismuth bicycles. Bicyclic peptides present a privileged molecular scaffold as the structure preorganization can seed high potency and specificity as potential therapeutics. Various methods have been developed in literature for the construction of peptide bicycles. The current contribution provides a unique and novel addition to bicyclic scaffolds as the bicyclization is accomplished through a single Bi atom, which yields perhaps the tightest peptide bicycles. Although the Bismuth bicyclization was previously reported, the current contribution optimizes the protocol (new bicyclization reagent) to allow construction of bismuth bicycle phage libraries. The resulting libraries were screened against streptavidin, which revealed peptide bicycles with single digit micromolar potency to streptavidin. This contribution has high novelty due to the unique peptide bicyclization. The streptavidin screen, although proof-of-concept, demonstrates the potential utility of the bismuth bicycles. I am delighted to support the publication of this work.

Reviewer #2 (Remarks to the Author):

Ullrich et al. present an interesting extension of their recently developed peptide bicyclisation strategy, applying this to phage display. In this manuscript they demonstrate that peptides displayed on a phage coat protein can be rapidly bicyclised using either bismuth or arsenic and the derived peptides used to identify binders of a model protein, streptavidin. This new cyclisation procedure appears simple and robust and I can see may find use by other researchers trying to develop targeted alpha therapies - following revision I am sure it will be of interest to the readers of Communication Chemistry. Overall I think the experimental work has been performed robustly and nice methods are included for most procedures. However, I do have some concerns with how the data has been presented which I believe should be addressed prior to publication.

While the introduction is well written I think it would benefit from a clearer introduction to the potential uses of bismuth containing peptides and the reasons for peptide bicyclisation. The authors describe peptide-bismuth bicycles as "an emerging class of constrained peptide". To support this they only cite two of their own recent papers - unless there are other related manuscripts I think this is a bit of an overreach. It would be better to explain the desirability of bismuth containing peptides and bicycles in general and then say that they've recently introduced ways to make them.

In a similar vein I think the issues with the TBMB chemistry have been overblown. This is used extensively in industry and academia for peptide discovery to great effect. The current introduction gives the impression that there are significant challenges using TBMB with phage which I don't think there are.

More significantly, I think the data presentation has been rather too neatly curated to fit the story and would benefit from a less cherry-picked presentation in the final manuscript.

For example, why have only peptides 1,3,4 been given for the BiBr3 library in Fig 2b and 1,3,6 in 2c. My assumption is that these were chosen because they contain the HPQ-like motif whereas peptide 2 for BiBr3 has an extra Cys and peptide 2 for gastrodenol doesn't have this HPQ motif (though it may well still bind)? It is not unreasonable to have synthesised the ones thought most likely to bind but I think Fig 2 should give a fair representation of the top hits and would be best showing e.g. the top 5 sequences from each screen. The text states that the (pg 4 line 116) arsenic screen enriched identical sequences but from my inspection of the SI sequencing data it looks as if they're pretty similar but not identical and in different orders, whilst pg 4 line 115 says they're distinct from the Bi screens but actually many of the sequences overlap - the top sequence in Fig S7 (Bi) is the 24th most enriched sequence in Fig S8 (As). Based on this I suspect many of the sequences will bind when cyclised with either As or Bi and this should be tested for at least one of the peptides if the authors want to make their assertion in the discussion (pg 8 178-183) that differences in geometry from the two chemistries leads to different peptide selectivity.

Minor comments:

Why was the pVIII linkage selected? Was this just because these were the libraries available to you? If so I think it would be better to display the conventional approach in Fig 1b as being either via pIII or pVIII so that the switch from pIII to pVIII doesn't appear to be a key feature of your method. Alternatively if you did select this for some reason it should be explained in the text.

Pg 4 line 112 as I understand it the two As(III) screens were exact replicates. It would be good to state this more clearly

Fig 2 – I find these dipeptide analyses rather hard to interpret and they are not really discussed in the text at all. If they are going to be included it would be good to explain what you think they show.

Pg 4 line 115/6 the sentence beginning "however" is grammatically unclear

Pg 6 line 124 – It would be good to link these sequences to sequences shown in Fig 2 and to give a sentence or two on why they were selected

Fig 3b – it would be good to be given the expected peptide masses and the assigned identities of the main adducts in these spectra

Methods:

Include statement in main text that more detailed methods are included in the SI

The authors do not purify the peptides after cyclisation with 1.1 fold excess of Bi/As. Is there any risk of this remaining reagent reacting with thiols in the target protein?

Reviewer #3 (Remarks to the Author):

The article provided new methods for bicyclisation of genetically encoded peptide libraries through bismuth or arsenic, realizing instantaneous, selective and entirely biocompatible modification of phage libraries. Through streptavidin screening campaigns, the authors identified enriched cyclic peptides with greater affinity compared to linear peptides, highlighting the effectiveness and reliability of the modified phage libraries in drug discovery. However, there is some weakness and need to be addressed in current form before publication.

1. As mentioned in this paper, previous researches disclosed that the bicyclisation of peptides through bismuth tribromide (BiBr₃) reacted instantaneously at room temperature and exhibited strong selectivity for cysteine residues. However, does it work for biological phage display? The authors need to demonstrate the real bicyclization of peptide libraries on bacteriophage. And additional experiments are needed to confirm the proportion of cyclized phage in 5 minutes, as described in Figure S1.
2. By using gastrodenol instead of BiBr₃, the authors aimed to avoid using DMSO and create a biocompatible bismuth bicyclic peptide library. As described in lines 103-104 "we conducted screenings with both bismuth reagents to assess their efficacy in phage display." It is essential to discuss the advantages of phage display with gastrodenol over BiBr₃, such as the capacity of libraries, the enriched sequences of screening and so on.
3. As mentioned in line 162, the impact of metal bismuth and arsenic on bacteriophage infective function should be evaluated, along with investigating the capacity of bicyclic peptide libraries after reaction. Although input phage concentrates of the naïve 3-cysteine phage library in all rounds

have been presented, the authors need to provide the phage inputs in actual biopanning after modification, which might have differences among the four libraries.

4. In Figure 4, there are significant fluctuations in the results of SPR experiments for 2b. The authors should consider conducting additional replicates for 2b and testing linear peptides multiple times to ensure result consistency. In addition, it would be good when add the data of positive control biotin for the reliability of experimental results.

5. The Nanopore sequencing data suggest that the HPQ/M motif only appears in the first CX4C loop of the bicyclic peptides, while the second loop does not exhibit this motif. Could the authors explain the reason for this occurrence?

6. In Figure S13, the LC-MS and ESI spectra of arsenic-peptide bicycles 3a and 4a should be included for a comprehensive presentation of the experimental findings.

Reviewer #2

While the introduction is well written I think it would benefit from a clearer introduction to the potential uses of bismuth containing peptides and the reasons for peptide bicyclisation. The authors describe peptide-bismuth bicycles as “an emerging class of constrained peptide”. To support this they only cite two of their own recent papers – unless there are other related manuscripts I think this is a bit of an overreach. It would be better to explain the desirability of bismuth containing peptides and bicycles in general and then say that they’ve recently introduced ways to make them.

We have adapted the style and flow of our introduction. The narrative now moves from cyclic peptides in the first paragraph to bicycles and the increased interest in them in the second paragraph.

In a similar vein I think the issues with the TBMB chemistry have been overblown. This is used extensively in industry and academia for peptide discovery to great effect. The current introduction gives the impression that there are significant challenges using TBMB with phage which I don’t think there are.

We have adjusted the paragraph on TBMB, where we now succinctly describe the limitations of this reagent in comparison to our method.

More significantly, I think the data presentation has been rather too neatly curated to fit the story and would benefit from a less cherry-picked presentation in the final manuscript. For example, why have only peptides 1,3,4 been given for the BiBr3 library in Fig 2b and 1,3,6 in 2c. My assumption is that these were chosen because they contain the HPQ-like motif whereas peptide 2 for BiBr3 has an extra Cys and peptide 2 for gastrodenol doesn’t have this HPQ motif (though it may well still bind)? It is not unreasonable to have synthesised the ones thought most likely to bind but I think Fig 2 should give a fair representation of the top hits and would be best showing e.g. the top 5 sequences from each screen.

We indeed select peptides based on their enrichment and occurrence of the HPQ/HPM motif from next-generation sequencing data which is common practice. We now show the top 6 sequences in Figure 2 as requested. To provide full transparency of the enrichment data, we now provide the analysed sequencing data as an additional supplementary file for the interested reader.

The text states that the (pg 4 line 116) arsenic screen enriched identical sequences but from my inspection of the SI sequencing data it looks as if they're pretty similar but not identical and in different orders, whilst pg 4 line 115 says they're distinct from the Bi screens but actually many of the sequences overlap – the top sequence in Fig S7 (Bi) is the 24th most enriched sequence in Fig S8 (As). Based on this I suspect many of the sequences will bind when cyclised with either As or Bi and this should be tested for at least one of the peptides if the authors want to make their assertion in the discussion (pg 8 178-183) that differences in geometry from the two chemistries leads to different peptide selectivity.

Following the reviewer's suggestion, we generated the bismuth variants of peptides **3** and **4** enriched in the arsenic screening. The additional data are discussed in the manuscript and included in Figure 4 and the Supporting Information.

Why was the pVIII linkage selected? Was this just because these were the libraries available to you? If so I think it would be better to display the conventional approach in Fig 1b as being either via pIII or pVIII so that the switch from pIII to pVIII doesn't appear to be a key feature of your method. Alternatively if you did select this for some reason it should be explained in the text.

The pVIII linkage was chosen because the pVIII library was indeed readily available to us. We adjusted Figure 1 and its caption to clarify that the choice of pIII or pVIII is not a fundamental requirement for neither the conventional nor our approach.

Pg 4 line 112 as I understand it the two As(III) screens were exact replicates. It would be good to state this more clearly

This is now stated as requested.

Fig 2 – I find these dipeptide analyses rather hard to interpret and they are not really discussed in the text at all. If they are going to be included it would be good to explain what you think they show.

This is now discussed in the main text as requested.

Pg 4 line 115/6 the sentence beginning “however” is grammatically unclear

This has been corrected and clarified.

Pg 6 line 124 – It would be good to link these sequences to sequences shown in Fig 2 and to give a sentence or two on why they were selected

This was added to Figure 2 as requested.

Fig 3b – it would be good to be given the expected peptide masses and the assigned identities of the main adducts in these spectra

In order to not overload the figure, we decided to include this information in an extra figure and table in the Supporting Information. This is also now discussed in the caption.

Methods: Include statement in main text that more detailed methods are included in the SI

This has been added as requested.

The authors do not purify the peptides after cyclisation with 1.1 fold excess of Bi/As. Is there any risk of this remaining reagent reacting with thiols in the target protein?

The target protein streptavidin does not contain any cysteines (thiols), hence no unspecific interactions are expected. While we see little reason for general concern about the 10% excess metal in this particular *in situ* modification and screening, protocols for the purification of the bismuth-peptide bicycles are available from our previous studies (Angew. Chem. Int. Ed. 2022 and 2024).

Reviewer #3

As mentioned in this paper, previous researches disclosed that the bicyclisation of peptides through bismuth tribromide (BiBr₃) reacted instantaneously at room temperature and exhibited strong selectivity for cysteine residues. However, does it work for biological phage display? The authors need to demonstrate the real bicyclization of peptide libraries on bacteriophage. And additional experiments are needed to confirm the proportion of cyclized phage in 5 minutes, as described in Figure S1.

The fact that we obtain bicyclic binders that are 100-fold more active than their linear congeners is a very strong indication that the modification works. In order to further address the reviewer's comment, we designed a model system using the soluble protein domain GB1 containing an N-terminal insert matching the ACX₄CX₄CGGGENLYFQS sequence from our phage libraries. We indeed observe quantitative modification of this model protein by bismuth within only 5 minutes of treatment with gastrodinol. The additional data are now included in the Supporting Information and discussed in the manuscript.

By using gastrodinol instead of BiBr₃, the authors aimed to avoid using DMSO and create a biocompatible bismuth bicyclic peptide library. As described in lines 103-104 "we conducted screenings with both bismuth reagents to assess their efficacy in phage display." It is essential to discuss the advantages of phage display with gastrodinol over BiBr₃, such as the capacity of libraries, the enriched sequences of screening and so on.

We have added a brief discussion to the manuscript as suggested.

As mentioned in line 162, the impact of metal bismuth and arsenic on bacteriophage infective function should be evaluated, along with investigating the capacity of bicyclic peptide libraries after reaction. Although input phage concentrates of the naïve 3-cysteine phage library in all rounds have been presented, the authors need to provide the phage inputs in actual biopanning after modification, which might have differences among the four libraries.

We have conducted additional experiments demonstrating that 5 min exposure to 120 μM of NaAsO₂, BiBr₃ or gastrodinol has no relevant negative effect on phage infectivity. This data has been added to the Supporting Information and is briefly discussed in the manuscript.

In Figure 4, there are significant fluctuations in the results of SPR experiments for 2b. The authors should consider conducting additional replicates for 2b and testing linear peptides multiple times to ensure result consistency. In addition, it would be good when add the data of positive control biotin for the reliability of experimental results.

It is customary to perform SPR experiments in a single measurement, given the robustness of the technique (e.g., ACS Cent. Sci., **7**, 1001–1008 or Nat. Chem., **15**, 998–1005). Nevertheless, for added assurance, we had already conducted the SPR analysis of our bicyclic peptides in duplicate. Concerning the perceived fluctuation in the results of bicyclic peptide **2b**, it is important to note that RU_{\max} appears different only due to the variable levels of immobilization within these two particular SPR channels. The critical parts of these two curves remain consistent, as evidenced by a standard deviation of the two dissociation constants being less than 1 μM . As for the inclusion of biotin as a control, we think that there is no additional benefit that would justify the additional experimental costs. The extraordinary affinity of biotin ($K_d = 10^{-15}$ M) would simply block the streptavidin channel on the SPR chip from further use.

The Nanopore sequencing data suggest that the HPQ/M motif only appears in the first CX4C loop of the bicyclic peptides, while the second loop does not exhibit this motif. Could the authors explain the reason for this occurrence?

We consider it possible that steric effects might be partly causing this effect. We now briefly discuss potential causes of this observation in the manuscript. We now also provide the full analysis of our Nanopore data as supplementary data, where HPQ/M sequences in the C-terminal loop occur but are certainly rarer.

In Figure S13, the LC-MS and ESI spectra of arsenic-peptide bicycles 3a and 4a should be included for a comprehensive presentation of the experimental findings.

We have now included the LC-MS data of all peptide bicycles in the Supporting Information as requested.

REVIEWERS' COMMENTS:

Reviewer #2 (Remarks to the Author):

I am happy that the authors have satisfactorily addressed my comments and those of the other reviewer.

Reviewer #3 (Remarks to the Author):

All the questions have been addressed by the authors. I have no further questions.